# Improving cardiovascular disease risk communication in NHS Health Checks: a qualitative study

Meredith K D Hawking,[1] Adam Timmis,[2] Fae Wilkins,[1] Jessica L Potter,[1] John Robson[1]

¹Centre for Primary Care and Public Health, Barts and The London School of Medicine and Dentistry, Queen Mary University of London, London, UK
²NIHR Cardiovascular Biomedical Research Unit, Barts Heart Centre, London, UK

**Correspondence to**
Dr John Robson;
j.robson@qmul.ac.uk

## ABSTRACT

**Objective** The NHS Health Check programme is a public screening and prevention initiative in England to detect early signs of cardiovascular ill health among healthy adults. We aimed to explore patient perspectives and experiences of a personalised Risk Report designed to improve cardiovascular risk communication in the NHS Health Check.

**Design and setting** This is a qualitative study with NHS Health Check attendees in three general practices in the London Borough of Newham.

**Intervention and participants** A personalised Risk Report for the NHS Health Check was developed to improve communication of results and advice. The Risk Report was embedded in the electronic health record, printed with auto-filled results and used as a discussion aid during the NHS Health Check, and was a take-home record of information and advice on risk reduction for the attendees. 18 purposively sampled socially diverse participants took part in semistructured interviews, which were analysed thematically.

**Results** For most participants, the NHS Health Check was an opportunity for reassurance and assessment, and the Risk Report was an enduring record that supported risk understanding, with impact beyond the individual. For a minority, ambivalence towards the Risk Report occurred in the context of attending for other reasons, and risk and lifestyle advice were not internalised or acted on.

**Conclusion** Our findings demonstrate the potential of a personalised Risk Report as a useful intervention in NHS Health Checks for enhancing patient understanding of cardiovascular risk and strategies for risk reduction. Also highlighted are the challenges that must be overcome to ensure transferability of these benefits to diverse patient groups.

**Trial registration number** NCT02486913.

## INTRODUCTION

The national NHS Health Check programme, initiated in 2009 across England, is a publicly funded screening and prevention programme aiming to detect early signs of cardiovascular ill health among healthy individuals aged 40–74 years old in the general population.[1] Effective communication of cardiovascular disease (CVD) risk is a core element of this programme, but previous qualitative research

## Strengths and limitations of this study

► This study was carried out during routine delivery of the NHS Health Check programme and included socially diverse participants at low or moderate cardiovascular risk.
► The study was limited to delivery in a general practice setting and may not be transferable to delivery in other settings.
► The Risk Report was only available in an English-language version and may be less suitable for people who prefer another language.

has suggested that patients struggle to understand risk in NHS Health Checks[2–4] and are dissatisfied with the lack of information provided.[5] A survey of patients found that over 70% recalled receiving lifestyle advice, but very few remembered receiving a CVD risk score and many incorrectly believed themselves to be low risk.[5] Research on attitudes of attendees and non-attendees called for consistent provision of tailored lifestyle information,[6] and cited limited communication of risk and inadequate access to support services as prime concerns relevant across differing age groups, ethnic groups and social groups.[7 8]

In East London the majority of NHS Health Checks are delivered by trained healthcare assistants (HCAs), typically multilingual staff drawn from local communities.[9] At present in East London there is no formal, standardised mechanism for conveying CVD risk information other than verbal communication during the NHS Health Check itself.

To improve communication in NHS Health Checks, we developed an evidence-based, personalised NHS Health Check Risk Report to be used both as a discussion aid and as an enduring record for patients to take away. This incorporated an infographic explanation of their CVD risk, as well as findings from clinical tests and a personalised action plan.

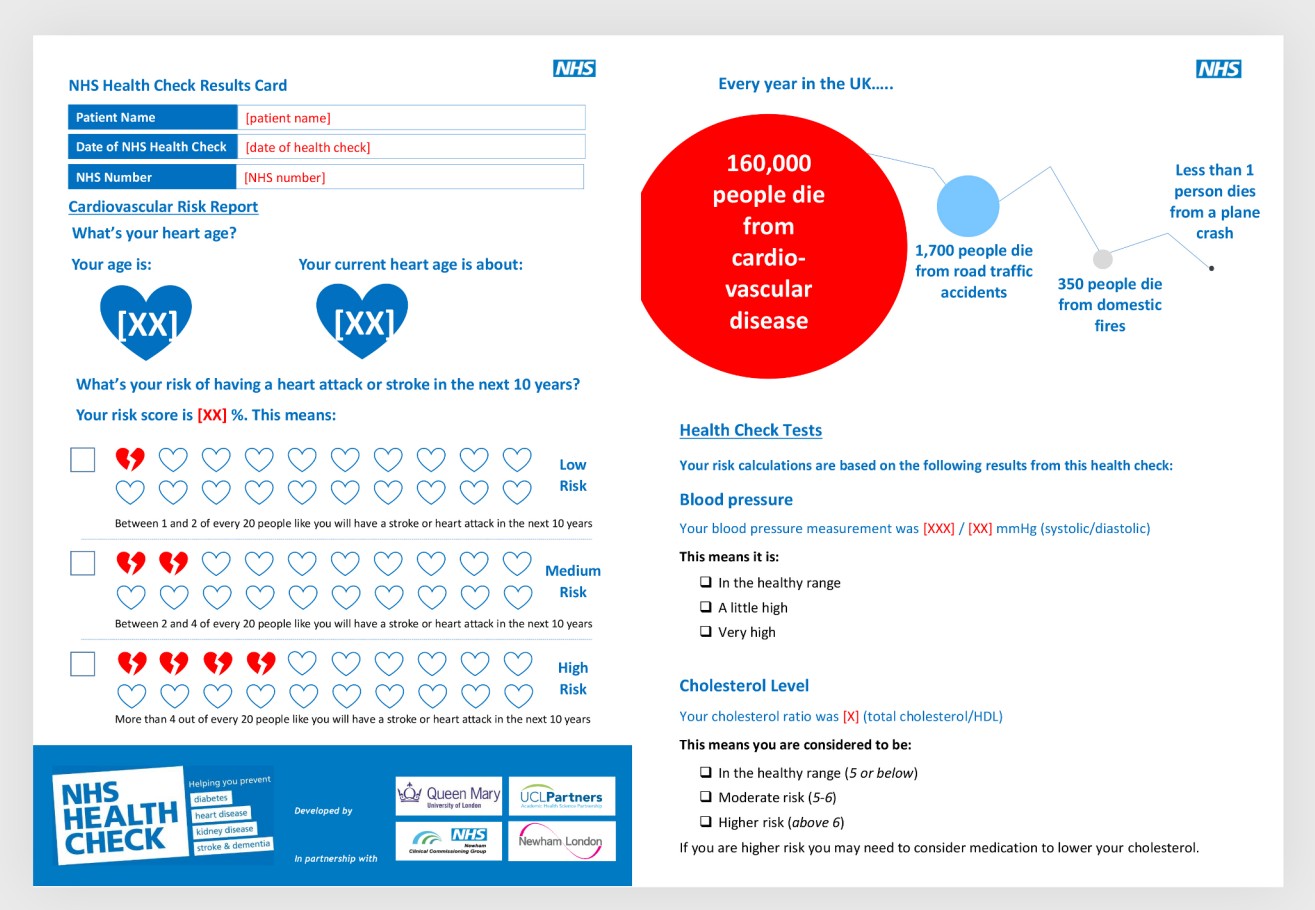

**Figure 1** Risk Report example pages.

This paper reports on qualitative findings from a mixed-methods feasibility trial of the Risk Report in general practice.

## METHODS
### Development of the Risk Report

The Risk Report (figure 1) was developed in line with national NHS Health Check programme guidelines and informed by the EAST (easy, attractive, social and timely) framework,[10] risk communication literature, and focus groups with HCAs, nurses and clinicians who are involved in the provision of Health Checks.

Input on implementation, content and design was gathered from general practitioners, specialist cardiologists and two informal focus groups with HCAs and nurses. Implementation barriers included limited access to colour printers, high printing costs and time taken to hand-write documents, and therefore most patients leave with no record of their results, the goals they have been set or resources for achieving them. In response, the Risk Report was designed to be embedded and saved within the electronic health record (EHR), to be automatically populated with patient-specific clinical information, easily printed off and discussed with the patient, and suitable for patients to take home.

### Public involvement

Patients/the public were not directly involved in the research planning or delivery process, including research question, recruitment, data collection or analysis. The design of the Risk Report was informed by the literature on client responses to information and risk communication and, as this was a feasibility study, modified in the light of responses by study participants. The qualitative interviews were designed to gain insight from participants into the delivery of improved methods of risk communication.

### Content of the Risk Report

We shaped elements of the Risk Report around the four principles of the EAST framework: easy, attractive, social and timely.[10]

► *Easy*: embedding the Risk Report in the EHR and simple printing cut 'hassle and time factors'. Complex behavioural goals were simplified by breaking down complex goals into achievable steps and including supporting resources.

► *Attractive*: National Health Service (NHS) branding was used to improve trustworthiness,[11 12] alongside infographics, personalisation and plain English wording. Health benefits of behaviour change were presented with alternative incentives to boost motivation to change.

- *Social*: social norms messaging was used to motivate behaviour change and increase salience. The action planning section includes family and friends in goal setting, prompts discussion and encourages making a commitment to others.
- *Timely*: delivery of messages is optimal when in the appointment setting, but continues as patients reflect on and refer back to the report. Messaging highlights immediate as well as long-term benefits of making behavioural changes.

The Risk Report includes QRISK2[13] and heart age[14] metrics to convey CVD risk, using infographics, icon arrays and pictorial natural frequencies with a common denominator to ease understanding and reduce denominator neglect.[15–18] Risk messages include a temporal component to provide context.[19] An infographic comparing comparative 'dread' risks is included to help patients situate their CVD risk alongside other causes of mortality.[20] We used survival framing to encourage risk-averse choices in terms of medication taking[21] and positive framing of messages to highlight alternative benefits of making healthy behaviours.[22] The Risk Report is printed in greyscale on four sides of A4 and includes information on local lifestyle change support services. It is available to view in full in online supplementary file.

## Study design

This is a nested qualitative interview study within a randomised feasibility trial. A completed Consolidated Criteria for Reporting Qualitative research checklist for this study is available in online supplementary file.

## Recruitment and selection

People aged 40–64 years due to be invited for an NHS Health Check were identified from six general practices in Newham, East London and were invited to attend two checks, 3–6 months apart. A list of 250 patients were randomly allocated to intervention or control groups. The study code denoting assignment group was then entered in the general practice record to identify these patients to the healthcare practitioners. The intervention group received a printed NHS Health Check Risk Report with a verbal explanation of its content at the first check, and the control group received usual care without written advice at the first check and the Risk Report with a verbal explanation of its content at the second check, following a waiting list control method. Patients in either group were eligible for the qualitative study after they had undertaken the two checks and received the Risk Report in either the first or second check.

Participants were recruited from three practices, out of six in the wider feasibility trial. Study information, consent form and an invitation to attend the NHS Health Check were provided in advance of recruitment. Participants gave written informed consent to take part in a qualitative interview at enrolment into the feasibility trial, and again verbally prior to interview commencement.

---

**Box 1  Topic guide**

- Early impressions of the NHS Health Check programme.
- Motivations to attend/accessibility.
- Overall experience.
- Key information/messages.
- Nurse/HCA communication.
- Understanding risk.

**Introduce Risk Report**

- General feedback (first impressions/comprehensiveness and so on).
- Design.
- Suggestions for improvement.
- Risk information.
- Lifestyle changes.
- Further comments.

HCA, healthcare assistant.

---

Participants were purposively sampled according to the information held in the EHR following a maximum diversity sampling approach, according to four categories. These were age, gender, ethnic group and CVD risk score (QRISK2[13]). QRISK2 score was categorised as low (a score of 10 or less), medium (score of 10–19) or high risk (score of 20 or more) according to the cut-offs used by the NHS Health Check programme. Ethnicity as reported within the medical record was grouped into the following categories: white (British, European), black (black (British/African or Caribbean)), South Asian (Indian, Pakistani, Bangladeshi, Sri Lankan) and all other ethnicities. Patients were telephoned and invited to participate by the project administrator or a practice receptionist.

## Data collection

The development of the topic guide (box 1) was based on a literature review and was piloted at one of the practices. The topic guide used open questions and probes to gather more details, and was adapted as new themes emerged. Single-instance semistructured interviews were undertaken in English, face to face in a private room at the practice by an experienced female qualitative researcher (MKDH) with a background in public health and health communication. Participants had no contact with the researcher before interview, and MKDH was introduced as a university researcher interested in improving NHS Health Checks. The Risk Report was introduced part way through the interview as a prompt and prop for discussion. Sessions lasted 10–40 min and were audio-recorded, pseudonymised and then transcribed. Reflective field notes were taken after each interview. Data collection occurred in March 2016–July 2017, as part of the wider trial which took place in March 2016–December 2017. Interviews commenced until saturation occurred and no new themes were arising.[23] Interviewees received a £20 shopping voucher for their time.

## Analysis

Inductive thematic analysis of the qualitative data was undertaken,[24] which involved coding the transcripts according to latent and manifest content and developing a thematic framework.[24] Analysis by MKDH proceeded alongside data collection and NVivo V.10 software was used for data management.[25] Of the transcripts 10% were double-coded by a second experienced qualitative researcher (JLP). Attention was paid to the development of narratives within and across transcripts. Any inconsistencies were discussed and agreed among the research team to offer multiple perspectives on the development of codes and identification of themes, and overarching thematic connections.

## RESULTS

### Sample characteristics

Eighteen respondents were well represented across gender, age group and self-defined ethnic group (table 1). Respondents at lower cardiovascular risk (QRISK2 score) were well represented, but those at high risk were not represented in the sample. Of those approached, none refused to take part in the interview.

### Overview of themes

Our thematic framework resulted in the identification of four main themes, which are outlined in table 2. On further interpretation we recognised two overarching threads in our data. These were patterns in the linking of themes. We discuss the findings within each theme and subtheme before discussing how the themes link together.

### Differing motivations

#### Detecting disease

Most participant accounts of why they attended the NHS Health Check were underscored by a desire to check their fitness or to screen for previously undetected disease, especially in the context of increasing age. This was framed as "maintaining your health."

> C101: Because I, I'm over 50 now, so I guess now is the right time to say you will have a check on your health, and try and see how fit you are.

There was a sense that some interviewees wanted to be reassured "just in case," with reported fears linked to having a family history of CVD or previous test results, such as high blood pressure results.

#### Being a willing patient

Among a minority of interviewees, the reason they attended was out of a sense of obligation to the practice and to the NHS more generally. One participant wanted to be seen as a "willing patient." Another attended to make sure they did not get removed from the practice register.

> C103: Probably to keep in with the practice, you know, showing that I was a good, energetic, willing patient.

**Table 1** Characteristics of the sample

| Group | Gender | | Age group (years) | | | QRISK2 score (%) | | | Ethnicity by grouping | | | |
|---|---|---|---|---|---|---|---|---|---|---|---|---|
| | Male | Female | 40–50 | 51–60 | 60–74 | 10 or less | 10–19 | 20 or more | White (British, European) | Black (British/African or Caribbean) | South Asian (Indian, Pakistani, Bangladeshi, Sri Lankan) | All other ethnicities |
| Control | 3 | 4 | 2 | 4 | 1 | 6 | 1 | 0 | 0 | 3 | 2 | 2 |
| Intervention | 8 | 3 | 2 | 5 | 4 | 5 | 6 | 0 | 2 | 5 | 4 | 0 |
| Total | 11 | 7 | 4 | 9 | 5 | 11 | 7 | 0 | 2 | 8 | 6 | 2 |

**Table 2** Thematic framework

| Differing motivations | Multiple risks | Risk Report as an enduring record | Impact |
|---|---|---|---|
| | Binary risk Risk meanings | | |
| Detecting disease | Acceptance of risk | 'Proof' of results | Making changes |
| Being a willing patient | Rejection of risk | Ambivalency | Not making changes |

## Multiple risks

There were multiple constructions of 'risk' in the interviews, reflecting the differences in participants' perceptions of their own health, whether they were concerned and whether or not there were language barriers during the health check.

### Binary risk

Those for whom English was not a first language mostly communicated their CVD risk in general, binary terms when asked, reporting that they were "fine" or "not at risk." Risk for these interviewees was either a state of being 'at risk' or not. Percentage risk and/or heart age were not mentioned or discussed.

### Risk meanings

Those that discussed risk in terms of their risk scores applied different meanings and importance to QRISK2 and heart age. For instance, this participant was explaining how she would like to have known her QRISK2 score because the two types of risk have different meanings for her:

> C202: This [heart age] is good to look at, it makes me feel young and things like, oh yeah I'm healthy-ish but our heart age doesn't, it's nice but I would have liked to have known my [Q] risk … I think it would make you make more changes when you can actually see, OK, so I'm really at risk here […] because even though it's 44 yeah, I don't know the risks.

Most participants were quicker to remember and report their heart age spontaneously in comparison with their QRISK2 score.

### Acceptance of risk

Interviewees reacted to their CVD risk scores in varied ways based on their perceptions of their state of health before attending the check. Those who mentioned being concerned about their CVD health prior to attending the NHS Health Check were more likely to report their risk score and accept it as a "true" reflection of their current state of health:

> C204: And she told me about the age, like my heart is 54 years old when I'm just 51. Yeah, you don't reach, I wanted it to be lower […] I'm going to try my best to bring it lower. That's what I'm trying to do right now.

### Rejection of risk

Some participants already believed themselves to lead a healthy life or have a healthy heart, and this meant that they disagreed with or discounted the CVD risk that was presented to them in the check if it did not fall in line with their own perceptions. For them, their risk was predefined by how they felt and saw themselves prior to the check:

> C103: Yeah, now that was a bit odd actually because the first time she said that there was a 1 in 10 chance of me having a heart attack and I thought, well that just doesn't seem right, once I got home I thought, no, that really doesn't seem right […] So that was a bit odd but I thought, I'm not going to have one [a CVD event] anyway. No. [laughs]

For these participants, their perception of not being "at risk" was based on feeling healthy and leading a healthy life, a lack of symptoms or the belief that good health is bestowed by a higher power.

## Risk Report as an enduring record

The Risk Report prompted participants to recall their CVD risk results. First impressions of the Risk Report were positive, as this example shows:

> C110: I thought this was quite interesting, straightaway got the message across, the risk levels and things.

Participants found it to be "user friendly" and clear with a good size font, no jargon and good for people for whom English was not their first language, or for those without a "scientific" background.

> C109: I'd just assume if my mother-in-law came and did this, this would be just right for her. Whereas, yeah if, if there was another person with a scientific background, we might suggest even more [information] but yeah, I think that was enough.

### 'Proof' of results

The majority of participants talked positively about the Risk Report as an enduring record of their results:

> C202: I think I liked the bit of paper that they gave me at the end that just jots down everything because I think you forget really easily and that's been good to look back at.

For some, this had additional meaning, as "proof" and reassurance of their good health, as this quote describes:

C109: Having something positive as proof on a piece of paper, like you could actually, a physical piece of paper. I know it sounds silly right but having a, it's like, hey if the pain is still there, like I say, at least my heart's working properly.

Participants reported looking back at the Risk Report to jog their memory. Some shared and discussed it with family and friends. Two reported keeping it by their bed, and some discussed how they would keep it to compare their results in future checks.

C207: Some leaflets you can look at and throw it away but when you look at that box and you're look at it and you're thinking, oh yeah that's interesting. […] Yeah, it does motivate you, as it has for me anyway, it really has.

### Ambivalency

A few interviewees were not interested in the Risk Report as a record at all, had not looked back at it or engaged with it:

I: Did you get a leaflet [showing Risk Report] like that?

C205: Might be. I haven't opened it since. It might still be in the house.

### Impact
#### Making changes

Most patients reported making at least small changes as a result of the NHS Health Check, such as incorporating more walks, reducing their salt and fat intake, and cutting down on alcohol consumption. A few participants had made significant changes—for instance, one patient had begun exercising regularly, joined a gym with his partner, cut down on fried foods, avoided drinking strong spirits and begun smoking less, and reported his cholesterol reduced from 5.5 to 4.9 in 3 months.

#### Not making changes

There were a few patients who had not made any changes to their lifestyle after the Check. These tended to be patients who reported that they were low risk or who already felt they were in good health. One patient, who had a medium QRISK2 score, was pre-diabetic and had been referred on for further tests, had tried to make changes but did not manage to keep it up:

C104: Well I want to but you've not got the go all the time, sometimes you just want to binge, what I do a lot. […] The diet plans, my son printed some sheets off from work, what you're supposed to be eating, but you'd do it for a few days and then you put them in a drawer.

Another patient had also been referred on for more tests to investigate his glucose levels but had not attended the referral clinic. He clearly understood his risk and what had been asked of him, however did not change his

behaviour according to the NHS Health Check results and recommendations:

C205: I was supposed to go for that check up […] And I didn't go for that. […] As long as I can still run and walk and I don't have pains and aches and dizzy spells as they like to call them. I don't. I don't worry.

### Wider impact

It was clear from the interviews that there was a wider social impact of NHS Health Checks that went beyond individuals. This included sharing the information from the Check with others, family members being included in diet and exercise changes, as well as recommendations to family and friends that they should attend their NHS Health Check.

C202: So now I have to look at my diet and actually even for my children as well, and look at their diet and just think, is it necessary, do we need all this salt, and I do really like salt and we are like a salty family. I would just add salt to everything and now I'm starting to think that I can't.

C207: I showed her [wife] and yeah and I've looked at it and it's when we go shopping now we, we probably have more vegetables and fruits in our shopping trolley than we have of all crisps.

### Bringing the themes together

The majority of participant accounts played out according to the main thematic findings described above. Participants attended their NHS Health Check to assess their levels of fitness and to get reassurance that nothing was wrong, or because they were concerned about previous tests or predisposition to ill health. They accepted the level of risk that the screening assigned to them, and viewed the Risk Report as a positive enduring record of their health status. They then went on to make lifestyle changes and share these with their families.

A minority of participant accounts in the study followed a different narrative from the central themes. These participants attended out of obligation or duty rather than concern for their health. They did not readily accept or internalise risk scores that differed from their own assessments of being healthy. They were ambivalent towards the Risk Report and did not make advised lifestyle changes. These diverse cases were few but are important for our consideration and recognition.

### DISCUSSION
#### Summary

This study sought to explore patient experiences of a personalised Risk Report designed to improve cardiovascular risk communication in the NHS Health Check. For most, the NHS Health Check was an opportunity for reassurance and assessment, and the Risk Report was an enduring record that supported risk understanding, and

supported lifestyle change for the individual and their wider social networks. For a minority, ambivalence towards the Risk Report occurred in the context of attending for other reasons, and risk and lifestyle advice were not internalised or acted on.

### Strengths and limitations

Our sample included medium and low CVD risk participants from a range of ethnic groups and ages, reflecting the Newham population, of whom the majority are low to medium risk.[9] Due to the targeted delivery approach adopted locally and high uptake rates in this area,[9] at the time of the study a large proportion of high-risk patients had already been identified and referred for further specialist services and support. In contrast, many low-risk to medium-risk patients are unlikely to receive further clinical referrals as a result of the NHS Health Check, and as such the Risk Report may be the only intervention that they receive. This group are therefore most likely to benefit from the action planning and further resources sections provided within the Risk Report, and are an appropriate target group for this intervention and as the focus of this study. The majority (16/18) of participants in this study were from black and ethnic minority backgrounds, which reflects in part the diversity of the population in Newham Borough. While this is an encouraging start, further adaptation may be required to tailor particular elements of the Risk Report for diverse population groups and cultural backgrounds.

Our findings may not be transferable to those who do not take up the offer of the NHS Health Check, nor to those who do not speak English. Participants were recruited from a wider feasibility trial, and so were patients who had both consented to the check and to take part in research. Participants were from both control and intervention groups, but, as both received the Risk Report either at the first check or the second (3 months apart), and patients were not aware which group they were allocated to, this may not have had a big impact on the data we collected. The inclusion of four patients for whom English was not their first language was pragmatic and reflects routine care in many areas, where healthcare professionals must communicate with patients about CVD risk regardless of language barriers and without the assistance of professional translators. Our finding that risk is constructed as 'binary' as a result of language barriers, even in the presence of a Risk Report with explanatory graphics, shows how risk is constructed in a fundamentally different way in comparison with numerical constructions of risk used by patients without those communication barriers. The impact this has on perceptions of health and health outcomes among this group warrants further investigation. Specific adaptations to the Risk Report and the NHS Health Check itself are required to extend the programme more equitably to all types of patient groups.

### Comparison with existing literature

Familial experience with CVD[26] and concerns about preventable or undetected illness as motivators to attend[27 28] have been reported elsewhere. Some participants in our study presented due to specific concerns, such as high blood pressure, supporting Perry et al's[4] assertion that it is not only the 'worried well' who attend. Burgess and colleagues[29] found civic responsibility to be a driver for some patients, which falls in line with being seen as a 'willing patient'. This study builds on these findings by aligning motivations to attend with the ongoing experiences of having a health check.

One key finding was the notion of CVD risk as a binary category, rather than a percentage scale. The 'take home message' from the NHS Health Check was often limited to dualistic generalities like being 'fine' or 'not at risk'. Whether people at higher CVD risk receive meaningful information and people at lower CVD risk are not falsely reassured should be a key focus of research in this area. The concept of an 'MOT', or 'roadworthiness'[30]—used prominently in the advertising of some NHS Checks—in which some people 'pass', and presumably need to take no further action, and the others 'fail' may not be the most effective message to convey. van Steenkiste and colleagues[31] found that communicated test results faded into an 'overall reassuring message' instead of specific results. Numerically, most heart attacks and strokes occur in people at moderate CVD risk, so if health checks are to be successful the focus must remain on the importance of reducing the burden of CVD in all patients, not just those at high risk. Challenging the 'MOT' metaphor associated with the overall programme may also help in this regard.

A key finding of our study was the ways in which patients used the Risk Report to motivate their own lifestyle changes, and as a way to prompt and support discussions and changes with others in their family and broader social networks. This finding has been mirrored in ethnographic work in a community health check setting, where Afro-Caribbean participants went on to discuss their results with peers.[26] In our study the 'work' of becoming healthier was a shared endeavour, involving those around the 'patient', including older relatives and children. Information impacted beyond the person through social practices such as eating and being physically active. Moving away from an individualistic approach to examine shared practices in the aftermath of health checks may be a fruitful area for future research.

The majority of participant views aligned with the dominant biomedical risk narrative behind screening for preventable illness in the UK. This viewpoint places the responsibility for maintaining health on the individual, via 'correct' lifestyle choices.[32] Having an NHS Health Check feeds this narrative by offering the opportunity for reassurance and provision of guidance to make small changes to maintain or protect health from uncertainty or risk. However, we found a minority of participants did not engage with the risk element of the NHS Health Checks, or intend to make lifestyle changes as a result of having a check, instead they attended for other reasons and were ambivalent about the Risk Report and the idea of 'improving' their health. These patients are often characterised as 'non-attenders' without an attempt to

explore their worldview.[33 34] Models of continuous care, typified by trusting clinician–patient relationships[35] and attention to patient narratives,[36] may allow for better ways of communicating about cardiovascular ill health with patients whose perspectives do not align with the dominant biomedical narrative. Additionally, qualitative attention to the whole patient and their journey through the NHS Health Check and beyond could help identify alternative ways of delivering good care for these groups of patients.

### Implications for research and practice

Our qualitative results support further development of a Risk Report for NHS Health Checks to enhance risk communication and support lifestyle modification and dissemination of messages among wider social networks. Our findings also highlight some of the challenges faced by the NHS Health Check programme for supporting patients whose ideas about risk and screening for future illness do not align with NHS and preventative public health priorities. Shifting the focus of the NHS Health Checks beyond the individual, to consider how networks of family and friends might interact, may have a wider positive influence on those wishing to lead healthier lives.

**Acknowledgements** We are grateful to the NHS Health Check attendees for giving up their time to take part in the study and share their experiences. We also extend our thanks for the cooperation and support from general practitioners and the clinic staff who were involved in delivering the study and those who helped inform the development of the Risk Report.

**Contributors** MKDH led the design of the Risk Report, and with AT and JR designed the qualitative study. FW organised the recruitment of practices, IT components, staff training and participant attendance. MKDH carried out the interviews and analysed the transcripts with JLP. All authors contributed to the manuscript.

**Funding** This study was funded by The Guttmann Academic Partnership hosted by UCLPartners. MKDH was in part supported by the National Institute for Health Research (NIHR) Collaboration for Leadership in Applied Health Research and Care North Thames at Barts Health NHS Trust (NIHR CLAHRC North Thames). The views expressed in this article are those of the author(s) and not necessarily those of the NHS, the NIHR, or the Department of Health and Social Care.

**Competing interests** JR has a non-pecuniary interest as a coauthor of QRISK2 and is also a member of a national NHS Health Check advisory group.

**Patient consent for publication** Not required.

**Ethics approval** The study received final approval from the National Research Ethics Service Committee North West (Preston) on 7 August 2015. The research ethics committee reference was 15/NW/0635 (protocol number MCPH1C8R).

**Provenance and peer review** Not commissioned; externally peer reviewed.

**Data availability statement** No data are available.

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
