## [Reviewer comments · BMJ Open]

ARTICLE DETAILS

TITLE (PROVISIONAL)	Improving cardiovascular disease risk communication in NHS Health Checks: a qualitative study
AUTHORS	Hawking, Meredith; Timmis, Adam; Wilkins, Fae; Potter, Jessica; Robson, John

VERSION 1 – REVIEW

REVIEWER	Željko Reiner University Hospital Center, Zagreb, Croatia
REVIEW RETURNED	02-Dec-2018

GENERAL COMMENTS	The topics of this manuscript is interesting. However, there are some major and some minor problems with it. The most important problem is that the whole manuscript is based upon the answers of only 18 people so the results cannot be representative for anything. This is the major limitation of this study and the authors did not even mention it, not to say discuss it, in the Discussion as a limitation. Actually, they do not mention any limitations of the study. Minor issue is that the authors should compare their observations with those in other similar studies performed in other countries and discuss this (with citations), for example with Reiner Z et al. Prev Med. 2010;51:494-496. and Reiner Z et al Atherosclerosis. 2010;213:598-603.
--

REVIEWER	Abdul Salam THE GEORGE INSTITUTE FOR GLOBAL HEALTH, India
REVIEW RETURNED	19-Dec-2018

GENERAL COMMENTS	Overall: The objective was "to explore patient perspectives and experiences of a personalised Risk Report, designed to improve cardiovascular risk communication in the NHS Health Check.". However, I think there is more information on NHS health check and its impact, patients perception of risk etc, rather than the the risk report it self. I think the paper would benefit with presenting more information on patient's feedback on the risk report. Reporting: overall reporting is good, referring to COREQ checklist would improve the presantation further. Page 4: the weblink to risk report is not working. I think better to provide it as an appendix/supplement to the paper. Page 5: line 9-10 check grammar
--

REVIEWER	Carolina Malta Hansen Emergency Medical Services Copenhagen, North Zealand Hospital, Division of Cardiology
REVIEW RETURNED	26-Feb-2019

GENERAL COMMENTS	This is a well-written paper describing a highly relevant study to explore patient perspectives and experiences of a personalised Risk Report, designed to improve cardiovascular risk communication in the NHS Health Check. This is a qualitative study using semi-structured interviews with a purposely sampled group of 18 patients. Thematic analysis was used to analyse the data. The main findings were that a personalised Risk Report is potentially a useful intervention in NHS Health Checks for enhancing patient understanding of cardiovascular risk and strategies for risk reduction. Challenges that must be overcome to ensure transferability of these benefits to diverse patient groups were also described. This is inspiring work and I would like to congratulate the team and authors on this intervention which serves as an example to improve the outreach of public health initiatives. Please consider the comments below which may further improve the manuscript:  1. One sentence explaining the NHS Health Check in the beginning of abstract would be helpful for readers not familiar with this intervention. It is well explained in the introduction but not sufficiently clear in the abstract. 2. Page 4, methods (content of risk report): it seems as though there is a Figure missing. The text says 'Insert Figure 1' but there is no Figure. 3. Page 4, methods (study design): 'Nested qualitative interview study within a randomised feasibility trial. The study methods for the trial have been reported separately (linked trial report).' To which trial are the authors referring? There is no link or reference. 4. Page 4: methods (recruitment and selection): it would be helpful to the reader to have insight into which ethnic groups were considered, and how you determined CVD risk. 5. Page 5, line 10: missing 'had' 6. It is not clear to the reader that you have a control and an intervention group. This should be described in the abstract and I guess it is what the authors mean by a randomized feasibility trial. But it was not clear to me until I reached the results section. What is the randomization? The report? Or another intervention? It is also not clear why you have 12 participants from the intervention and only 7 from the control group. Also, how are the results from the control group included in the study? This is confusing and needs to be clarified. 7. Results: There seems to be a mistake in Table 1: the total number of males is 11 but there are 9 in the intervention and 3 in the control group. 8. Results/methods: why were there no patients with a high risk score? These would seemingly be the ones who would benefit the most. 9. Discussion: it is usually helpful to the reader to begin the discussion with a statement of the purpose of the study: ' This study which sought to explore patient experiences.... Etc.' 10. Discussion: The authors mention the Risk Report may be the only intervention low and intermediate patients receive. This is a
--

	relevant point, which the authors do not explore further. How could this intervention then benefit this low/intermediate risk population? And how are primary physicians then supposed to use this tool to guide patient information and treatment? 11. Discussion: How did the report perform among the 4 patients who were not native speakers? Any thoughts on that? Any thoughts on how different cultural backgrounds may react to such a report?
--	--

VERSION 1 – AUTHOR RESPONSE

Reviewer: 1

Reviewer Name: Željko Reiner

Institution and Country: University Hospital Center, Zagreb, Croatia

Please state any competing interests or state 'None declared': None declared

Please leave your comments for the authors below

The topic of this manuscript is interesting. However, there are some major and some minor problems with it. The most important problem is that the whole manuscript is based upon the answers of only 18 people so the results cannot be representative for anything. This is the major limitation of this study and the authors did not even mention it, not to say discuss it, in the discussion as a limitation.

Author's response: This study was a qualitative study, using semi-structured interviews as a data collection methodology, analysed thematically. 18 participants in this kind of study is acceptable when justified and guided by concepts of thematic saturation (1) and information power (2). In this case, we recruited patients according to purposive sampling categories until thematic saturation was achieved. Although we acknowledge that there is some debate about the use of 'saturation' as a concept within qualitative research (3), we politely disagree with this comment and do not see the sample size as a limitation of the study.

Actually, they do not mention any limitations of the study.

Author's response: Limitations of the study that we discuss in the paper include: issues of language in data collection, limitations of transferability of the findings due to representativeness of the sample in terms of patients with high CVD risk, non-English language speakers, and patients who do not take up the offer of the health check.

Minor issue is that the authors should compare their observations with those in other similar studies performed in other countries and discuss this (with citations), for example with Reiner Z et al. *Prev Med.* 2010;51:494-496. and Reiner Z et al *Atherosclerosis.* 2010;213:598-603.

Author's response: Within our discussion we have discussed and compared our observations with a number of similar qualitative studies in similar 'health check' contexts, for instance those undertaken by: Riley et al 2015, Ellis et al 2015, Jenkinson et al 2015, Burgess et al 2015, Shaw et al 2015, van Steenkiste et al 2004, Dalton et al 2011, Dryden et al 2012. Thank you for bringing our attention to these research papers which describe large cross sectional surveys assessing perceptions of CVD risk amongst the general public attending pharmacies, and general practitioners in Croatia. This method and population context differs from our study as the findings are not linked to the delivery of an intervention or according to provision of routine health checks or screening. We weren't able to address broader patterns of CVD risk perception amongst the public because qualitative methodology provides insight into contextually specific, situated phenomenon, rather than results that are generalizable across contexts.

Reviewer: 2

Reviewer Name: Abdul Salam

Institution and Country: THE GEORGE INSTITUTE FOR GLOBAL HEALTH, India

Please state any competing interests or state 'None declared': None declared

Please leave your comments for the authors below

Overall: The objective was "to explore patient perspectives and experiences of a personalised Risk Report, designed to improve cardiovascular risk communication in the NHS Health Check". However, I think there is more information on NHS health check and its impact, patients' perception of risk etc., rather than the risk report itself. I think the paper would benefit with presenting more information on patient's feedback on the risk report.

Author's response: Thank you for your comment. The patient perception of risk in the context of NHS Health Check is inextricably tied to the risk report, as this is the way that the risk was communicated and constructed in this particular situated context. However we accept your feedback and have added some further detail to the summarised findings in the third theme 'risk report as an enduring record' on page 9, which explicitly addresses feedback on the risk report.

Reporting: overall reporting is good, referring to COREQ checklist would improve the presentation further.

Author's response: Thank you for pointing this out. We did use the COREQ checklist, which was submitted as an additional file alongside our paper, and have now indicated this within the methods section (page 4) as suggested here.

Page 4: the web link to risk report is not working. I think better to provide it as an appendix/supplement to the paper.

Author's response: we have now submitted the risk report as a supplementary file as you have suggested, and indicated this on page 4.

Page 5: line 9-10 check grammar

Author's response: thanks, we have now added in the missing word 'had' here.

Reviewer: 3

Reviewer Name: Carolina Malta Hansen

Institution and Country: Emergency Medical Services Copenhagen, North Zealand Hospital, Division of Cardiology

Please state any competing interests or state 'None declared': None Declared

Please leave your comments for the authors below

Please see the file attached.

This is a well-written paper describing a highly relevant study to explore patient perspectives and experiences of a personalised Risk Report, designed to improve cardiovascular risk communication in the NHS Health Check. This is a qualitative study using semi-structured interviews with a purposely sampled group of 18 patients. Thematic analysis was used to analyse the data. The main findings were that a personalised Risk Report is potentially a useful intervention in NHS Health Checks for

enhancing patient understanding of cardiovascular risk and strategies for risk reduction. Challenges that must be overcome to ensure transferability of these benefits to diverse patient groups were also described.

This is inspiring work and I would like to congratulate the team and authors on this intervention which serves as an example to improve the outreach of public health initiatives. Please consider the comments below which may further improve the manuscript:

1. One sentence explaining the NHS Health Check in the beginning of abstract would be helpful for readers not familiar with this intervention. It is well explained in the introduction but not sufficiently clear in the abstract.

Author's response: Thank you for this feedback. We have now included a sentence in the objective section of the abstract on page 2.

2. Page 4, methods (content of risk report): it seems as though there is a Figure missing. The text says 'Insert Figure 1' but there is no Figure.

Author's response: Thank you for bringing this to our attention - this was an Authors' note reminding us to link to the uploaded high resolution figure of the risk report, and this note has now been deleted.

3. Page 4, methods (study design): 'Nested qualitative interview study within a randomised feasibility trial. The study methods for the trial have been reported separately (linked trial report).' To which trial are the authors referring? There is no link or reference.

Author's response: When we submitted this paper, we also simultaneously submitted another linked paper to the same journal that has not been accepted for publication at this time. We agree this is confusing and we have therefore included further relevant details in the manuscript about the wider feasibility trial (methods section, on page 4/5), including addressing the issues raised below in comment 6.

4. Page 4: methods (recruitment and selection): it would be helpful to the reader to have insight into which ethnic groups were considered, and how you determined CVD risk.

Author's response: CVD risk for purposive sampling was assessed using the QRISK2 score (4) as recorded in the electronic health record, split into low (a score of 10 or less), medium (score: 10-19) and high risk (score of 20 or more) according to the cut offs used by the NHS Health Check programme. There was no limit in terms of ethnic group, we used the categories as defined within the practice medical records. We have now included more information about these details in the methods section on page 5.

5. Page 5, line 10: missing 'had'

Author's response: thank you for bringing this to our attention - we have added in the missing word.

6. It is not clear to the reader that you have a control and an intervention group. This should be described in the abstract and I guess it is what the authors mean by a randomized feasibility trial. But it was not clear to me until I reached the results section. What is the randomization? The report? Or another intervention? It is also not clear why you have 12 participants from the intervention and only 7 from the control group. Also, how are the results from the control group included in the study? This is confusing and needs to be clarified.

Author's response: Agreed – please see our response to comment 3. As both control and intervention participants received the risk report in one of their health checks (we used a waiting list control), we were able to include both groups within the qualitative study. However, we have added a comment on this in the limitations section of the discussion (page 12).

7. Results: There seems to be a mistake in Table 1: the total number of males is 11 but there are 9 in the intervention and 3 in the control group.

Author's response: Thank you for spotting this – we have now corrected this typo on page 6.

8. Results/methods: why were there no patients with a high risk score? These would seemingly be the ones who would benefit the most.

Author's response: Due to the targeted delivery approach adopted in the local area and high uptake rates, at the time of the study a large proportion of high risk patients had already been identified and referred for further specialist services and support. The pool of potential participants in this group was therefore very low. These patients, however, go on to be referred for extra support which goes

beyond what the risk report offers in terms of behavioural and clinical intervention. In the amendments to the manuscript I have tied discussion of this point with point 10 below – please see the updated sections on page 10/11 of the manuscript.

9. Discussion: it is usually helpful to the reader to begin the discussion with a statement of the purpose of the study: 'This study which sought to explore patient experiences.... Etc.'

Author's response: Thank you - we have now added in a statement on the purpose of the study at the beginning of the discussion – page 10.

10. Discussion: The authors mention the Risk Report may be the only intervention low and intermediate patients receive. This is a relevant point, which the authors do not explore further. How could this intervention then benefit this low/intermediate risk population? And how are primary physicians then supposed to use this tool to guide patient information and treatment?

Author's response: We have elaborated on this point in the discussion section on pages 10/11 in the manuscript.

11. Discussion: How did the report perform among the 4 patients who were not native speakers? Any thoughts on that?

Author's response: I have added some reflection on page 12 about how language barriers altered how risk was constructed as binary amongst non-native speakers, in spite of the numerical information included in the report. This was such a small group so I would hesitate to draw any broader conclusions about language, but what is clear is that further work would be necessary to 'translate' and adapt the report for this group.

Any thoughts on how different cultural backgrounds may react to such a report?

Author's response: We had a diverse sample, with the majority of the participants from black and ethnic minority groups (16/18 participants), which is a strength of the study. Our findings therefore reflect a diverse range of different cultural backgrounds. However, this was a feasibility study and therefore cultural adaptation of the risk report may be required in the future, although more work would be required to tailor the information appropriately, which was beyond the scope of this early study. I have added some detail about this on page 12.

References

1. Mason M. Sample Size and Saturation in PhD Studies Using Qualitative Interviews. 2010. 2010;11(3).
2. Malterud K, Siersma VD, Guassora AD. Sample Size in Qualitative Interview Studies: Guided by Information Power. Qual Health Res. 2016;26(13):1753-60.
3. O'Reilly M, Parker N. 'Unsatisfactory Saturation': a critical exploration of the notion of saturated sample sizes in qualitative research. Qualitative Research. 2012;13(2):190-7.
4. Hippisley-Cox J, Coupland C, Vinogradova Y, Robson J, Minhas R, Sheikh A, et al. Predicting cardiovascular risk in England and Wales: prospective derivation and validation of QRISK2. BMJ. 2008;336(7659):1475.

VERSION 2 – REVIEW

REVIEWER	Carolina Malta Hansen Emergency Medical Services, Copenhagen, Copenhagen University Division of Cardiology, North Zealand Hospital, Capital Region of Denmark
REVIEW RETURNED	26-Mar-2019
GENERAL COMMENTS	No further comments.